# The Prognostic Role of Diagnostic Criteria for COVID-19-Associated Pulmonary Aspergillosis: A Cross-Sectional Retrospective Study

**DOI:** 10.3390/antibiotics13020150

**Published:** 2024-02-03

**Authors:** Erika Asperges, Rebecca Pesare, Cecilia Bassoli, Matteo Calia, Sonia Lerta, Francesco Citiolo, Giuseppe Albi, Caterina Cavanna, Paolo Sacchi, Raffaele Bruno

**Affiliations:** 1S.C. Malattie Infettive I, Fondazione IRCCS Policlinico San Matteo, 27100 Pavia, Italy; rebecca.pesare@gmail.com (R.P.); cecilia.bassoli01@universitadipavia.it (C.B.); matteo.calia01@universitadipavia.it (M.C.); sonia.lerta01@universitadipavia.it (S.L.); p.sacchi@smatteo.pv.it (P.S.); raffaele.bruno@unipv.it (R.B.); 2Department of Clinical-Surgical, Diagnostic, and Pediatric Sciences, University of Pavia, 27100 Pavia, Italy; francesco.citiolo01@universitadipavia.it; 3Department of Electrical, Computer and Biomedical Engineering, University of Pavia, 27100 Pavia, Italy; giuseppe.albi01@universitadipavia.it; 4Microbiology and Virology Department, IRCCS Policlinico San Matteo, 27100 Pavia, Italy; c.cavanna@smatteo.pv.it

**Keywords:** COVID-19, pulmonary aspergillosis, case definitions, CAPA, certain/probable/possible diagnosis

## Abstract

Several criteria exist to diagnose pulmonary aspergillosis with varying degrees of certainty in specific populations, including oncohaematological patients (EORTC/MSG), ICU patients (mAspICU) and COVID-19 patients (ECMM). At the beginning of the pandemic, however, the diagnosis of COVID-19-Associated Pulmonary Aspergillosis (CAPA) could not be performed easily, and the decision to treat (DTT) was empirical. In this cross-sectional retrospective study including patients with SARS-CoV-2 infection and suspicion of CAPA, we studied the concordance between the DTT and the three diagnostic criteria using Cohen’s coefficient, and then we identified the factors associated with the DTT and corrected them by treatment to study the influence of the diagnostic criteria on survival. We showed good concordance of the DTT and mAspICU and ECMM criteria, with “compatible signs”, “positive culture” and “positive galactomannan” influencing the DTT. Treatment also showed a positive effect on survival once corrected for a putative, possible or probable diagnosis of CAPA using mAspICU and ECMM criteria. We conclude that EORTC/MSGERC are not considered applicable in clinical practice due to the lack of inclusion of signs and symptoms and do not lead to improved survival. mAspICU and ECMM criteria showed a good degree of agreement with the DTT and a positive correlation with patient recovery.

## 1. Introduction

COVID-19-Associated Pulmonary Aspergillosis (CAPA) is an opportunistic infection associated with severe COVID-19. It was recognised as a clinical entity soon after the beginning of the pandemic, with a wildly varying incidence (5.7–35.0%) [1,2,3] and prevalence (3.0–33.3%) [4,5] depending on the diagnostic criteria.

Specific criteria for the diagnosis of invasive aspergillosis have existed since 2002, when the first European Organization for Research and Treatment of Cancer/Invasive Fungal Infections Cooperative Group and National Institute of Allergy and Infectious Diseases Mycoses Study Group (EORTC/MSG) criteria were compiled. The original article and its revisions propose three levels of certainty in the diagnosis of invasive aspergillosis (proven, probable and possible), and their intended use is aimed at research, not clinical practice [6,7,8]. Moreover, they focus on immunosuppressed patients (solid organ and haematopoietic stem-cell transplant recipients and solid cancer and haematological patients), with only the “proven” category being applicable to everyone, and thus their applicability to intensive care unit (ICU) and COVID-19 patients is limited [9].

Since then, efforts have been made to either expand these criteria to ICU patients [9] or to create new ones, like the AspICU algorithm [10], which has a more clinical focus and includes critically ill patients. The AspICU criteria distinguish patients with proven, putative pulmonary aspergillosis and colonisation. Later, Schauwvlieghe et al. modified the criteria (“modified AspICU”; mAspICU going forward), excluding the host factors (neutropenia, oncohaematological malignancy, glucocorticoid treatment and congenital or acquired immunosuppression) [11]. Their aim was to avoid the automatic exclusion from a diagnosis of invasive aspergillosis of critically ill patients with influenza, who rarely have host factors but are nonetheless at high risk of aspergillosis [12]. Since then, Influenza Associated Pulmonary Aspergillosis (IAPA) has been recognised as a clinical entity with its own dignity.

Finally, the European Confederation of Medical Mycology (ECMM) provided criteria specific to COVID-19 patients in 2020 [13]. The ECMM criteria are specifically targeted at both research and clinical guidance and were derived from EORTC/MSG and AspICU criteria. SARS-CoV-2 positivity and need for intensive care were considered entry criteria, and aspergillosis diagnosis was divided into proven, probable and possible.

mAspICU and ECMM criteria might seem similar at first glance since the main difference is COVID-19 as an entry factor in ECMM. However, ECMM also added factors specific to aspergillar tracheobronchitis, simplified the clinical factors and mandated slightly more specificity on the microbiological factors. Moreover, the entry factor “need for intensive care” of ECMM is different from the entry factor of “ICU patient” of mAspICU: it allows more flexibility, given that COVID-19 patients that required intensive care at the beginning of the pandemic were not necessarily admitted to the ICU due to bed unavailability, and consequently, they were often treated in ordinary or sub-intensive wards.

A summary of definitions of EORTC/MSG, mAspICU and ECMM criteria is provided in Appendix A, while a comparison of probable, possible, putative and colonisation diagnoses is provided in Appendix A.

Especially at the beginning of the pandemic, when specific diagnostic criteria were not available, and procedures like lung biopsy and bronchoalveolar lavage (BAL) were limited, the decision to treat (DTT) was often empirical or based on knowledge derived from older diagnostic algorithms (EORTC/MSG and AspICU/mAspICU) despite suspicions of their inadequacy. The main aim of our study is to retrospectively assess the concordance between DTT and the three criteria (EORTC/MSG, mAspICU and ECMM). While other criteria have been developed in the meantime (for example, BM-aspICU [14]), these have been widely validated and disseminated and are likely to be used frequently in clinical practice. We also aim to identify which of their factors are associated with the DTT. We then correct by treatment to study the influence of the diagnostic criteria on survival.

## 2. Results

### 2.1. Population

As described in Table 1, we retrospectively collected 196 patients with COVID-19 and a suspicion of CAPA. The median age was 64 years old (53–71). There were 139 male and 57 female patients. The median length of hospital stay was 26.1 days (14.3–49.0). Death was recorded for 66 patients (33.7%). ICU admission was recorded for 52 patients (44.8%), and mechanical ventilation (invasive and non-invasive) was needed for 65 patients (57.0%).

#### 2.1.1. mAspICU and ECMM Criteria

According to the mAspICU criteria, aspergillosis was certain in 0% of patients and putative in 18.0%. Colonisation was diagnosed in 1.8% of patients.

According to the ECMM criteria, aspergillosis was certain in 0% of patients, probable in 17.1% and possible in 1.8%.

The most prevalent host factors (applied as described in paragraph 4.4 and Table 1) were “use of steroids” (87.5%) and “presence of compatible signs” (74.1%).

#### 2.1.2. EORTC/MSG Criteria

According to the criteria, aspergillosis was certain in 0% of patients, probable in 9.0% and possible in 24.3%. The most prevalent host factor was the use of steroids (32.7%).

#### 2.1.3. Common Factors and Treatment

The main radiological pattern was atypical (83.8%), and galactomannan positivity was achieved in only 14.8% of patients. Aspergillus was cultured in 34 patients (30.4%). The DTT was applied in 33 patients (16.8%), mainly with voriconazole alone (24 patients, 12.2%).

### 2.2. Concordance of Diagnostic Criteria with the Decision to Treat

The 2 × 2 tables featuring the DTT and each of the three diagnostic criteria are shown in Table 2. Concordance among the DTT and the criteria is shown in Table 3. Cohen’s κ coefficient is 0.46 and 0.44 for mAspICU and ECMM criteria, respectively, showing good concordance with the DTT (*p* < 0.001 for both analyses), while it is only 0.16 for EORTC/MSG criteria (*p* > 0.05). A further comparison between mAspICU and ECMM criteria showed almost complete concordance (κ = 0.91, *p* < 0.001).

#### Factors Influencing the Decision to Treat

The logistic regression on criteria and their factors influencing the DTT was performed on 112 patients (who had available data on criteria factors and on treatment). The univariate analysis showed a significant influence of factors “compatible signs” and “positive galactomannan” (Table 4).

### 2.3. Correlation between Survival and Diagnostic Criteria

The logistic regression on criteria and their factors influencing survival was performed on 112 patients. The univariate analysis (Table 5) showed a significant association of mortality with cytotoxic agents and B-cell suppressants therapy. However, as shown in Table 6, when adjusted for the three diagnostic criteria (and cytotoxic agents and B-cell suppressants), treatment showed a significant association with survival when using mAspICU and ECMM criteria (OR 6.17 and 5.36, respectively; *p* < 0.05) but not EORTC/MSG (OR 3.80, *p* = 0.05).

## 3. Discussion

In our cohort of 196 patients, we found a prevalence of probable/possible/putative CAPA of 8.9–24.1% according to the criteria. No certain cases of CAPA were found; we attribute this to the difficulties in performing invasive procedures such as lung biopsy during the pandemic and in critically ill patients. The long median length of hospital stays, as well as the high rates of mechanical ventilation, ICU admission and death, were expected, given that our population included patients with severe COVID-19 before any knowledge on treatment or management became available. The prevalence of the male sex was also expected, as it is a known risk factor for severe COVID-19. Moreover, biological differences between males and females in immune response to fungal infections have been hypothesised and might have played a role [15].

We found a good agreement between the physicians’ DTT and the classification of patients according to mAspICU and ECMM diagnostic criteria. The factors that led to the DTT were the “presence of compatible signs” and “positive galactomannan”.

Both “compatible signs” and “positive galactomannan” are used by the mAspICU and ECMM criteria but not by the EORTC/MSG criteria, which, in fact, did not show concordance with the DTT. Moreover, the mAspICU and ECMM criteria showed almost perfect concordance between them.

Concerning survival, treatment showed a positive effect that was confirmed once corrected for a putative, possible or probable diagnosis of CAPA using mAspICU and ECMM criteria but not using EORTC/MSG.

The inappropriateness of the EORTC/MSG criteria for the diagnosis of CAPA has been suspected since the start of the pandemic. This was reasonable since the EORTC/MSG criteria were developed specifically for oncohaematological patients, while the AspICU/mAspICU and ECMM criteria were destined for ICU and COVID-19 patients. This inappropriateness has also been confirmed by the only other study we could find that compared the criteria for the diagnosis of CAPA [16]. The study was conducted in Germany on 684 critically ill patients and compared EORTC/MSG and mAspICU criteria, finding a Cohen’s κ of 0.14, which is comparable to our finding. Unfortunately, the analysis did not include ECMM criteria.

The concordance between mAspICU and ECMM criteria was less expected. In fact, the first reports of CAPA based on AspICU or mAspICU criteria led to an overestimation of its prevalence (up to 30%) [1,17], and even though larger studies brought this number down to 3.8% [18], they still advocated for a low threshold of suspicion, with the consequent risk of overtreatment. The development of the ECMM criteria had the effect of reducing the prevalence despite being based mostly on expert opinion and low-grade evidence [19], but variable incidence and prevalence are still reported [4,20]. Our work showed that the choice of starting a treatment does not rely on those factors that differ between the two criteria (the host factors) but on clinical presentation and microbiological factors. In fact, even though the microbiological factors of the ECMM are more stringent than those of mAspICU, the difference is minimal. This restricts the disagreement in management that would derive from the use of one diagnostic criterion over the other. This is further confirmed by our findings on survival: if treatment had been based on mAspICU or ECMM classification, it would have led to improved odds of survival (OR > 5, *p* < 0.05). The use of EORTC/MSG, on the other hand, would have still led to improved survival, with the OR being 3.80 and the lower limit of the confidence interval being >1, but it did not reach statistical significance.

A possible explanation of the inadequacy of the EORTC/MSG criteria lies in the restrictive nature of their host factors, which leave out not only non-oncohaematologial causes of immune suppression (for example, chronic inflammatory bowel diseases) but also other alterations of the immune system function that could still impact on the risk of opportunistic infections, like the use of anti-interleukin drugs. In this regard, tocilizumab is noteworthy because of its use in COVID-19. However, we point out that while tocilizumab might have an impact on the development of invasive aspergillosis, our results were not impacted by its use because only one patient in our cohort received it.

We did not find any literature on the effect of criteria on the DTT or their correlation to survival, although a Belgian study conducted on ICU patients found that, when CAPA is diagnosed with ECMM criteria, the additional presence of EORTC/MSG host factors leads to increased mortality [21].

Our study presents several limitations: apart from its single-centre and retrospective nature, the risk of selection bias is apparent in our assumption that patients with a suspicion of CAPA would have undergone at least one galactomannan antigen test. Another bias is our definition of a patient requiring the ICU, but since this study includes the first waves of the pandemic, where ICU beds were scarcely available and ordinary wards functioned as ICU units [22], we believe that this did not overly affect the results. In fact, patients needing non-invasive mechanical ventilation are still often treated in ordinary wards thanks to the skills acquired during the pandemic.

Prospective international studies on the impact of diagnostic criteria on accuracy and patients’ outcomes should be planned.

## 4. Materials and Methods

### 4.1. Study Design and Setting

We conducted a cross-sectional retrospective study including patients admitted to Fondazione IRCCS Policlinico San Matteo of Pavia from 21 February 2020 to 30 April 2022. All patients were hospitalised for a PCR-proven SARS-CoV-2 infection and were suspected of having CAPA.

### 4.2. Demographic and Clinical Data

Demographic and clinical data were retrospectively extracted from electronic medical records. The demographic data included age and gender; clinical data included wards of stay, date of admission and discharge, host risk factors for immunosuppressive condition according to aspergillosis diagnostic criteria, other relevant comorbidities, the use of invasive or non-invasive mechanical ventilation and clinical outcome. Microbiological data included fungal cultures and galactomannan (GM) antigen results from any sample. Chest X-rays or computed tomography (CT) results were reported as “normal” or with “typical” and “atypical” alterations. Antifungal therapy choice was also collected. Ninety-one patients had at least one missing value.

### 4.3. Laboratory and Radiological Investigations

Regarding laboratory investigations, we considered any galactomannan index ≥0.5 positive, regardless of a biological sample, in accordance with the manufacturer’s recommendations. This cutoff was suggested as the best compromise between specificity and sensitivity, with a relatively higher cutoff level (≥0.7) recommended only in special populations, such as patients receiving fungus-derived antibiotics or children receiving antineoplastic chemotherapy [23]. For assignation to the criteria, however, we used the cut-offs suggested by the criteria themselves (see Appendix A. For example, serum galactomannan ≥1.0 for probable aspergillosis according to EORTC/MSG or serum galactomannan ≥0.5 for putative aspergillosis according to mAspICU).

Culture testing for *Aspergillus* species was performed in all patients in whom representative material from the lower respiratory tract could be collected.

The radiological definitions were derived from the criteria themselves (see Appendix A). Typical radiological patterns, evidenced on X-ray or CT-scan examinations, include the following: dense, well-circumscribed lesion(s) with or without halo sign; air crescent sign; a cavity; or segmental or lobar consolidation. These typical signs are uncommon in non-neutropenic patients. Atypical radiological patterns refer to the presence of elements like unspecific infiltrates, diffuse ground-glass opacity and consolidations.

### 4.4. Cases’ Definitions According to Criteria

Patients were classified as cases of CAPA according to EORTC/MSG, mAspICU and ECMM criteria with a few modifications:-EORTC/MSG host factors related to an immunosuppressive condition [8] were applied as standard, but we equated prolonged use of dexamethasone or methylprednisolone prescribed for SARS-CoV-2 pneumonia to prolonged, high-dose of prednisone (≥0.3 mg/kg for more than 3 weeks).-Among the AspICU criteria, we chose the modified AspICU (mAspICU) [11], applied as standard. However, the entry factor “ICU admission” was also considered present in patients receiving mechanical ventilation in ordinary wards since these patients would have been admitted to the ICU prior to the pandemic.-ECMM criteria were applied faithfully [13].

### 4.5. Statistical Analysis

The patients’ clinical numerical characteristics were presented as median and interquartile ranges after the Shapiro test excluded the normal hypothesis, while categorical variables were presented with frequencies and percentages.

The concordance analyses between the three criteria and the DTT (defined as the physician’s decision to assign an antifungal treatment to the patient) and the agreement between the criteria themselves required the first step of making the criteria binary. Patients who could not be assigned to a diagnostic category within a given criterion (for example, for lack of the entry factor) or who were assigned to the “colonisation” category were given a value of 0, while patients assigned to the possible, probable or putative categories were given a value of 1. We then built 2 × 2 tables with the binary categories and the DTT. From these tables, we calculated Cohen’s κ coefficient and its *p*-value with the z-test. Cohen’s κ coefficient is used to quantify agreement between two evaluators. Similar to correlation coefficients, κ ranges from −1 to +1, where 0 expresses the amount of agreement that can be expected from random chance, a κ value < 0 can be interpreted as no agreement, while a value >0 indicates agreement. Positive values > 0.4 are usually interpreted as good concordance and >0.8 as excellent concordance.

We then evaluated with a univariate analysis the differences between the factors used to compile the diagnostic criteria, after dividing the study cohort according to the DTT. We used the non-parametric Mann–Whitney test to compare the numerical variables and the Chi-square or Fisher’s exact test, when appropriate, to compare the categorical variables binarised in 0/1 values. The same univariate analysis was repeated for the outcome of “survival”.

Finally, we assessed via a multivariate analysis the possible association between survival and the use of mAspICU, ECMM and EORTC/MSG criteria. We used logistic regression and included the criteria as predictors and the DTT variable as an additional confounding factor. Results of the multivariable analysis are reported as odds ratio (OR) and 95% confidence interval.

All the statistical tests were conducted two-tails, and a *p*-value < 0.05 was considered significant. Rstudio 4.0.5. with R version 4.1.2 was used for all the computation and statistical analysis.

### 4.6. Informed Consent and Ethical Concerns

All patients provided informed consent for the use of clinical data for scientific purposes according to hospital policy. Administrative data about our hospital’s COVID-19 patients are collected in a registry (SMACORE) approved by the Fondazione IRCCS Policlinico San Matteo’s ethics committee with protocol number 20200046877.

## 5. Conclusions

The evidence gathered in this study demonstrates that mAspICU and ECMM criteria have a good degree of agreement with the DTT and between themselves. Moreover, they show a positive correlation with patient recovery, further validating their use in daily clinical practice.

On the other hand, the use of EORTC/MSGERC criteria to guide the diagnosis is not considered applicable in clinical practice by physicians due to the lack of inclusion of clinical signs and symptoms. When antifungal therapy is prescribed according to its categories, it does not lead to improved survival.

## Figures and Tables

**Table 1 antibiotics-13-00150-t001:** Population description. Continuous data are presented with median and interquartile ranges (1°–3° quartile) and categorical data with frequency and percentages. ICU: intensive care unit; NA: not available; CPAP: continuous positive air pressure; BAL: Bronchoalveolar lavage.

Hospitalisation Registry (n = 196)	
Age [years]	64 (53–71)
Sex	
Male	139 (70.9)
Female	57 (29.1)
Outcome	
Deceased	66 (33.7)
Discharged	130 (66.3)
Length of stay [days]	26.1 (14.3–49.0)
ICU admission (NA: 80)	52 (44.8)
Mechanical ventilation (NA: 82)	
CPAP	18 (15.8)
Invasive	47 (41.2)
Comorbidities	
Diabetes (NA: 83)	20 (17.7)
Obesity (NA: 83)	11 (9.7)
Solid tumour (NA: 84)	13 (11.6)
Chronic Obstructive Pulmonary Disease (NA: 83)	24 (21.2)
mAspICU and ECMM factors	
Compatible signs (NA: 84)	83 (74.1)
Neutropenia: <0.5 × 10^9^ [neu/L] before or at ICU admission (NA: 81)	7 (6.1)
Cytotoxic agents (NA: 84)	16 (14.3)
Steroids: 20 mg/day (NA: 83)	98 (87.5)
Immunodeficiency (NA: 83)	11 (9.7)
EORTC/MSG host factors	
Neutropenia: <0.5 × 10^9^ [neu/L] for >10 days (NA: 80)	7 (6.0)
Haematological malignancy	25 (21.7)
Haematopoietic stem-cell transplantation (NA: 83)	11 (9.7)
Solid organ transplant	9 (7.8)
Steroids: 0.3 mg/kg/day for >3 weeks (NA: 83)	37 (32.6)
T-immunosuppressant (NA: 83)	12 (10.6)
Treatment with B-cell suppressors	4 (3.5)
Acute graft-versus-host disease	1 (0.9)
Inherited severe immunodeficiency (NA: 83)	1 (0.9)
Radiological pattern (NA: 91)	
Atypical	88 (83.8)
Normal	12 (11.4)
Typical	5 (4.8)
Galactomannan antigen	
Material (NA: 8)	
BAL	80 (42.6)
Serum	108 (57.4)
Positivity (NA: 7)	28 (14.8)
Discrepancy BAL/serum positivity	5 (2.6)
Median value (NA: 7)	0.2 (0.2–2)
Cultures	
Material (NA: 88)	
Bronchonasopharyngeal aspirate	6 (5.5)
Sputum	10 (9.3)
Induced sputum	1 (0.9)
Blood	1 (0.9)
BAL	90 (83.3)
Species (NA: 85)	
Aspergillus	17 (15.2)
Candida	58 (51.8)
*Aspergillus* + *Candida*	17 (15.2)
mAspICU classification (NA: 85)	
Certain	0 (0.0)
Putative	20 (18.0)
Colonisation	2 (1.8)
ECMM classification (NA: 85)	
Certain	0 (0.0)
Probable	19 (17.1)
Possible	2 (1.8)
EORTC/MSG classification (NA: 84)	
Certain	0 (0.0)
Probable	10 (8.9)
Possible	27 (24.1)
Treatment	
Decision to treat (NA: 86)	33 (16.8)
Drug	
Echinocandin	3 (6.5)
Fluconazole	2 (4.3)
Voriconazole	24 (12.2)
Isavuconazole	2 (4.3)
Voriconazole + amphotericin B	1 (2.2)

**Table 2 antibiotics-13-00150-t002:** The 2 × 2 tables showing the relationship between the decision to treat (DTT) and the classification according to mAspICU, ECMM and EORTC/MSG diagnostic criteria, simplified to binary criteria (patients to whom criteria are not applicable or are colonised; patients who have or might have pulmonary aspergillosis).

**mAspICU**			
		**Colonisation/Not Applicable**	**Certain/Putative**
Treatment	No	74	4
	Yes	16	13
**ECMM**			
		**Not Applicable**	**Certain/Probable/Possible**
Treatment	No	73	5
	Yes	16	13
**EORTC/MSG**			
		**Not Applicable**	**Certain/Probable/Possible**
Treatment	No	57	22
	Yes	16	13

**Table 3 antibiotics-13-00150-t003:** Concordance between DTT and mAspICU, ECMM and EORTC/MSG criteria expressed as Cohen’s κ coefficient. Green represents a concordance value with *p* < 0.001; red represents a concordance value with *p* > 0.05.

Concordance (Expressed as Cohen’s κ Coefficient)				
	DTT	mAspICU	ECMM	EORTC/MSG
DTT		0.46	0.44	0.16
mAspICU			0.91	0.02
ECMM				0.00
EORTC/MSG				

**Table 4 antibiotics-13-00150-t004:** Univariate analysis showing correlation between the criteria’s factors and the decision to treat. Significant *p*-values are in bold. DTT: decision to treat; NA: not available; ICU: intensive care unit.

Factors	DTT Yes (n = 31)	DTT No (n = 81)	*p* Value
Neutropenia: <0.5 × 10^9^ [neu/L] for >10 days	3 (9.7)	4 (4.9)	0.39
Neutropenia: <0.5 × 10^9^ [neu/L] before or at ICU admission	2 (6.5)	5 (6.2)	1
Haematological malignancy	8 (25.8)	16 (19.8)	0.66
Haematopoietic stem-cell transplantation (NA: 1)	5 (16.1)	6 (7.4)	0.17
Solid organ transplant	2 (6.5)	7 (8.6)	1
Steroids: 0.3 mg/kg/day for >3 weeks (NA: 1)	11 (35.5)	25 (32.1)	0.82
Steroids: 20 mg/day (NA: 2)	26 (90.3)	69 (87.3)	0.91
T-immunosuppressant (NA: 1)	5 (16.1)	7 (8.6)	0.42
B-immunosuppressant	-	4 (4.9)	0.57
Acute graft-versus-host disease	-	1 (1.2)	1
Inherited severe immunodeficiency (NA: 1)	1 (3.2)	-	0.27
Compatible signs (NA: 2)	28 (90.3)	54 (67.5)	**0.03**
Cytotoxic agents (NA: 1)	3 (10.0)	13 (16.2)	0.55
Immunodeficiency (NA: 1)	2 (6.5)	9 (11.1)	0.72
Atypical Radiologic pattern (NA: 9)	5 (16.7)	12 (16.7)	1
Galactomannan positivity	10 (32.2)	7 (8.6)	**<0.001**
Positive culture of Aspergillus (NA: 40)	17 (63.0)	25 (56.8)	0.79

**Table 5 antibiotics-13-00150-t005:** Univariate analysis showing correlation between survival and the diagnostic criteria’s factors. Significant *p*-values are in bold. NA: not available; ICU: intensive care unit.

Factors	Survivors (n = 86)	Deceased (n = 26)	*p* Value
Neutropenia: <0.5 × 10^9^ [neu/L] for >10 days	5 (5.8)	2 (7.7)	0.66
Neutropenia: <0.5 × 10^9^ [neu/L] before or at ICU admission	5 (5.8)	2 (7.7)	0.66
Haematological malignancy	16 (18.6)	8 (30.8)	0.29
Haematopoietic stem-cell transplantation	8 (9.3)	3 (11.5)	0.71
Solid organ transplant	8 (9.3)	1 (3.8)	0.68
Steroids: 0.3 mg/kg/day for >3 weeks	26 (30.2)	11 (42.3)	0.36
Steroids: 20 mg/day (NA: 2)	75 (87.2)	22 (91.7)	0.81
T-immunosuppressant	11 (12.8)	1 (3.8)	0.29
B-immunosuppressant	1 (1.2)	3 (11.5)	**0.04**
Acute graft-versus-host disease	1 (1.2)	-	1
Inherited severe immunodeficiency	1 (1.2)	-	1
Compatible signs (NA: 1)	63 (73.3)	19 (76.0)	0.99
Cytotoxic agents (NA: 1)	7 (8.2)	9 (34.6)	**0.002**
Immunodeficiency	7 (8.1)	4 (15.4)	0.28
Atypical Radiologic pattern (NA: 10)	16 (20.0)	1 (4.5)	0.11
Galactomannan positivity	12 (14.0)	5 (19.2)	0.73
Positive culture of Aspergillus (NA = 41)	30 (55.6)	12 (70.6)	0.41
Treatment (NA: 2)	26 (31.0)	3 (12.5)	0.12
mAspICU (NA: 1)	14 (16.3)	6 (24.0)	0.56
ECMM (NA: 1)	15 (17.4)	6 (24.0)	0.65
EORTC/MSG	25 (29.1)	12 (46.2)	0.16

**Table 6 antibiotics-13-00150-t006:** Multivariate analysis showing correlation between survival and diagnostic criteria. Significant *p*-values are in bold. OR: odds ratio.

	OR	95% Confidence Interval	*p* Value
All criteria			
Cytotoxic agents	0.18	0.04–0.76	**0.019**
B-cell suppressants	0.66	0.03–8.59	0.76
Treatment	7.12	1.67–43.71	**0.016**
mAspICU	0.42	0.01–7.04	0.564
ECMM	0.45	0.03–13.95	0.583
EORTC/MSG	0.53	0.18–1.59	0.243
mAspICU			
Cytotoxic agents	0.17	0.04–0.62	**0.008**
B-cell suppressants	0.58	0.02–7.03	0.687
Treatment	6.17	1.49–36.81	**0.023**
mAspICU	0.22	0.05–0.89	**0.035**
ECMM			
Cytotoxic agents	0.14	0.03–0.57	**0.006**
B-cell suppressants	0.64	0.03–7.80	0.738
Treatment	5.36	1.40–28.33	**0.025**
ECMM	0.23	0.06–0.92	**0.036**
EORTC/MSG			
Cytotoxic agents	0.25	0.06–0.97	**0.042**
B-cell suppressants	0.48	0.03–5.33	0.572
Treatment	3.80	1.10–18.16	0.054
EORTC	0.61	0.22–1.76	0.343

## Data Availability

The data presented in this study are available upon request from the corresponding author.

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
