# Peer review of "The Prognostic Role of Diagnostic Criteria for COVID-19-Associated Pulmonary Aspergillosis: A Cross-Sectional Retrospective Study"

_antibiotics, 2024, doi:10.3390/antibiotics13020150_

Round 1

Reviewer 1 Report

Comments and Suggestions for Authors

Dear authors, I have reviewed your paper and have the following comments:

MAJOR:

-it is not clear why only the three score presented in the paper have been used. For instance, the aspICU score has an update, the BMaspICU score, which is not mentioned but could have been more appropriate could look into. Please explain in the paper.

-in the results section, it is said that " ICU admission was recorded for 52 patients (44.8%) and mechanical ventilation (invasive and noninvasive) was needed for 65 patients (57.0%)". This is not in accordance with one of the tables in the paper and also does not make sense. Please review and amend.

-the galactomannan threshold mentioned seems very low. The cutoff for negativity is normally less than 60-80 pg/mL (and for BAL around 0.8 or even 1). This needs to be thoroughly explained.

-this type of statistical analysis is not standard for many clinicians, so maybe give a little bit more details about the Cohen coefficient. 

-are the figures quoted here: " with wildly varying incidence and prevalence depending on the diagnostic criteria (5,7%-27,7%)" incidence or prevalence figures? Cannot be both at the same time.

-how do you explain the fact that the M:F ratio is around 2:1? Anything worth discussing?

-the conclusion is not really new (this is why for CAPA another score was developed I would think, clinicians knew EORTC was not applicable) and is very succint. Try to elaborate a bit more.

MINOR:

-please amend "galattomannan" to galactomannan throughout the paper

Author Response

Dear reviewer, thank you for you comments. The revisions implemented after your suggestions undoubtedly improve our manuscript. Here are our responses: 

-it is not clear why only the three score presented in the paper have been used. For instance, the aspICU score has an update, the BMaspICU score, which is not mentioned but could have been more appropriate could look into. Please explain in the paper. We chose the three scores that are most commonly known and are validated and widely used. Regarding the aspICU score, we actually used one of the updated versions (the modified aspICU). This is specified in paragraph 4.4. The BM-aspICU has only rarely been used (for example, a Medline search only provides three results, one of which is the original study and another a case report) thus we chose not to include it. Moreover, to our knowledge, it was developed on a very small number of patients. However, we added a sentence to explain our choice in the Introduction: "While other criteria have been developed in the meantime (for example BM-aspICU [14]) these have been widely validated and disseminated, and are likely to be used frequently in clinical practice".  

-in the results section, it is said that " ICU admission was recorded for 52 patients (44.8%) and mechanical ventilation (invasive and noninvasive) was needed for 65 patients (57.0%)". This is not in accordance with one of the tables in the paper and also does not make sense. Please review and amend. The percentages are computed on the total of the patients for which the data is available. To clarify this, we added in the tables the number of missing data. 

-the galactomannan threshold mentioned seems very low. The cutoff for negativity is normally less than 60-80 pg/mL (and for BAL around 0.8 or even 1). This needs to be thoroughly explained. The cutoff was chosen according to the manufacturer’s recommendations. Moreover, several articles have confirmed 0.5 as an ideal cutoff in terms of sensitivity and specificity. We explained our choice and we have cited the review including these articles in paragraph 4.3. We also added that, apart from this, patients were assigned to the categories in three criteria using the definitions provided by the criteria themselves, which often, though not always, used 0.5.  

-this type of statistical analysis is not standard for many clinicians, so maybe give a little bit more details about the Cohen coefficient. We added details about why it is used and on its interpretation in paragraph 4.5: "Cohen's κ coefficient is used to quantify agreement between two evaluators. Similar to correlation coefficients, κ ranges from -1 to +1, where 0 expresses the amount of agreement that can be expected from random chance, a κ values <0 can be interpreted as no agreement, while a value >0 indicates agreement. Positive values >0.4 are usually interpreted as good concordance, >0.8 as excellent concordance."

-are the figures quoted here: " with wildly varying incidence and prevalence depending on the diagnostic criteria (5,7%-27,7%)" incidence or prevalence figures? Cannot be both at the same time. We fixed the sentences dividing incidence and prevalence. 

-how do you explain the fact that the M:F ratio is around 2:1? Anything worth discussing? This probably reflects the fact that male sex is a known risk factor for severe Covid-19. Moreover, biological differences between males and females in immunity response to fungal infections have been hypothesized. Two sentences about this have been added to the discussion: "The prevalence of the male sex was also expected, as it is a known risk factor for severe Covid-19. Moreover, biological differences between males and females in immune response to fungal infections have been hypothesized and might have played a role [15]."

-the conclusion is not really new (this is why for CAPA another score was developed I would think, clinicians knew EORTC was not applicable) and is very succint. Try to elaborate a bit more. We agree that the unsuitability of EORTC criteria was apparent from the start. However, as the most widely known criteria, they were still used for studies and to inform clinical decision making. The new information of our work is in demonstrating exactly what makes them untrustworthy in daily practice (lack of clinical and microbiological factors).We also add what we think is valuable information in demonstrating the concordance between AspICU and ECMM criteria and demonstrating their impact on survival. All of this is discussed in paragraph 3; the conclusion itself (paragraph 5) is just a summary of the previous observations, however we have reworded it to highlight what is new in our study. 

MINOR:

-please amend "galattomannan" to galactomannan throughout the paper: amended. 

Reviewer 2 Report

Comments and Suggestions for Authors

This interesting paper is an overview of CAPA diagnostic criteria comparing the three definitions of pulmonary aspergillosis in 196 patients with Covid-19 and suspected CAPA.

Here are my observations:

a. The three definitions of pulmonary aspergillosis (EORTC/MSG, AspICU and ECMM) should be presented clearly in a Supplement for the readers not confortable with their differences. Furthermore, the use of a comparing table with the probable and possible diagnosis would be helpful.

b. The three tables (n. 1, n. 4, and n. 5) of unusual length (longer than a page) are making difficult to go smoothly through the article. I would consider an abridged version of each table putting the unabridged table in the Supplements.

c. In the paragraph “Materials and Methods – Demographic and clinical data” it is stated that the microbiological data included even Beta-D-Glucan, but - then - no results are given about this meaningful parameter. Moreover, Galattomannan antigen is done in the serum and in the BAL without showing the concodance or discrepancy between serum and BAL once you present a positive result.

d. Again, in the paragraph “Materials and Methods – Laboratory and radiological investigations” it is stated that the typical radiological patterns of pulmonary aspergillosis are uncommon in non-neutropenic patients. This is generally accepted for air-crescent and halo sign, but not for the nodular lesions without halo and the cavities which are usually present in solid transplantation and other immunodeficiencies even if these are not patognomonic findings. Thereafter, in the results only the atypical pattern (unspecific infiltrates, diffuse ground glass opacity and consolidations) are reported and found pointless. Pulmonary nodular lesions should not be indicated among unspecific infiltrates despite not having patognomonic significance.

e. Data about Tocilizumab therapy are not reported despite this being a risk factor for opportunistic infections. A remark would be welcome.

f. Last but not least, a comment would be appreciated about the small clinical difference between AspICU and ECMM diagnostic criteria since almost all patients with Covid-19 and suspected pulmonary aspergillosis are located in the ICU.

Author Response

This interesting paper is an overview of CAPA diagnostic criteria comparing the three definitions of pulmonary aspergillosis in 196 patients with Covid-19 and suspected CAPA.

Thank you for the encouragement. The paper is now better thank to your suggestions. 

Here are my observations:

a. The three definitions of pulmonary aspergillosis (EORTC/MSG, AspICU and ECMM) should be presented clearly in a Supplement for the readers not confortable with their differences. Furthermore, the use of a comparing table with the probable and possible diagnosis would be helpful. The definitions and a comparing table have been provided in the Supplementary material as requested, and a sentence has been added in the introduction to refer to them. 

b. The three tables (n. 1, n. 4, and n. 5) of unusual length (longer than a page) are making difficult to go smoothly through the article. I would consider an abridged version of each table putting the unabridged table in the Supplements. We thank you for the suggestion, we will discuss this with the editor. 

c. In the paragraph “Materials and Methods – Demographic and clinical data” it is stated that the microbiological data included even Beta-D-Glucan, but - then - no results are given about this meaningful parameter. The data was collected but ultimately not analysed because it is not included in any of the three criteria. We have removed all references to it to avoid confusion. 

Moreover, Galattomannan antigen is done in the serum and in the BAL without showing the concodance or discrepancy between serum and BAL once you present a positive result. A line was added to table 1 to make explicit the number of patient with a discrepancy in serum/BAL galactomannan (5 patients). 

d. Again, in the paragraph “Materials and Methods – Laboratory and radiological investigations” it is stated that the typical radiological patterns of pulmonary aspergillosis are uncommon in non-neutropenic patients. This is generally accepted for air-crescent and halo sign, but not for the nodular lesions without halo and the cavities which are usually present in solid transplantation and other immunodeficiencies even if these are not patognomonic findings. Thereafter, in the results only the atypical pattern (unspecific infiltrates, diffuse ground glass opacity and consolidations) are reported and found pointless. Pulmonary nodular lesions should not be indicated among unspecific infiltrates despite not having patognomonic significance. While we understand the reviewer’s concerns, the definition of atypical radiological pattern is part of the criteria’s definition. Thus, the inclusion of nodular lesions as unspecific infiltrates is not dependent on us. We clarified this in paragraph 4.3 with a new sentence and a reference to the supplementary table. 

e. Data about Tocilizumab therapy are not reported despite this being a risk factor for opportunistic infections. A remark would be welcome. Data on treatment was available for 120 patients. Of these, only one received Tocilizumab, which is why we did not study it further. However, we agree with you on the importance of it as a risk factor and we now added a comment in the discussion: "A possible explanation of the inadequacy of EORTC/MSG criteria lies in the restrictive nature of their host factors, which leave out not only non-oncohematologial causes of immune-suppression (for example chronic inflammatory bowel diseases) but also other alterations of the immune system function that could still impact on the risk of opportunistic infections, like the use of anti-interleukin drugs. In this regard, tocilizumab is of noteworthy because of its use in Covid-19. However, while it might have an impact of the development of invasive aspergillosis, our results were not impacted by its use since only one patient received it. "

f. Last but not least, a comment would be appreciated about the small clinical difference between AspICU and ECMM diagnostic criteria since almost all patients with Covid-19 and suspected pulmonary aspergillosis are located in the ICU. A sub-paragraph that explains the differences between the two criteria has been added in the introduction: "mAspICU and ECMM criteria might seem similar, at first glance, since the main difference is Covid-19 as entry factor in ECMM. However, ECMM also added factors specific for aspergillar tracheobronchitis, simplified the clinical factors and mandated slightly more specificity on the microbiological factors. Moreover, the entry factor “need for intensive care” of ECMM is different from the entry factor of “ICU patient” of mAspICU: it allows more flexibility, given that Covid-19 patients that required intensive care at the beginning of the pandemic were not necessarily admitted to the ICU due to bed unavailability, and consequently they were often treated in ordinary or sub-intensive wards." We also added a further comment in the discussion: "Our work showed that the choice of starting a treatment does not rely on those factors that differ between the two criteria (the host factors) but on clinical presentation and microbiological factors. In fact, even though the microbiological factors of the ECMM are more stringent than those of mAspICU, the difference is minimal." 

Round 2

Reviewer 1 Report

Comments and Suggestions for Authors

No further comments